

# The physiological response of Ectomycorrhizal fungus *Lepista sordida* to Cd and Cu stress

Yin Dachuan and Qi Jinyu

College of Forestry, Shenyang Agricultural University, ShenYang, People's Republic of China

## ABSTRACT

Ectomycorrhizal fungi (ECMF) can develop the resistance of host plants to heavy metal stress. However, little is known about the response of ECMF to heavy metal exposure. In this study, the growth and physiological indices of *Lepista sordida* under Cd and Cu stress were studied. The growth of *L. sordida* on PDA medium under Cd and Cu stress was observed using scanning electron microscopy (SEM). After the addition of Cd and Cu to the medium, the mycelium started twisting, breaking, sticking together, and even dissolving. In the control group, a good and luxuriant mycelium growth of *L. sordida* along with the numerous clamp connections was observed. The mycelial biomass decreased with increasing concentrations of heavy metals in a liquid medium. The catalase (CAT), peroxidase (POD), superoxide dismutase (SOD), and ascorbate peroxidase (APX) activities were also investigated, and the results showed that the Cd and Cu treatments caused a significant increase in the antioxidant enzyme activities. The contents of soluble protein, soluble sugar, and free proline in *L. sordida* were investigated, and it was found that the contents initially increased and then decreased with the increasing concentrations of Cd and Cu. However, the content of malondialdehyde (MDA) increased with the increasing concentrations of Cd and Cu. In conclusion, the present study provides a theoretical basis for the better utilization of Ectomycorrhizal fungal resources for the remediation of soil contaminated with heavy metal.

# INTRODUCTION

Since the beginning of the 21st century, there has been a rapid increase in industrial development and production. This has also been accompanied by the unreasonable disposal of household waste. Heavy metal contamination of soil has become a severe environmental problem worldwide (*Ruttens et al., 2006*), attracting much attention from the scientific community (*Granero & Domingo, 2002*). The rapid industrialization of mining, smelting, lead batteries, and other industries, and the use of sewage to irrigate farmland have led to serious heavy metal contamination in the soil. Cadmium (Cd) and copper (Cu) are the most common heavy metal contaminants. Cd contamination exceeded the standard by 7.0 percent, ranking Cd first among all the pollutants (*Liu, Wu & Zhang, 2019*). Pollution of Cd and Cu in the soil leads to excessive levels of heavy metals in the soil, with the pollution

Corresponding author
Yin Dachuan, yin-dachuan@syau.edu.cn

of Cd and Cu in agricultural and forestry production increasingly at serious levels (*Wang et al., 2020*). Heavy metal pollution in the soil is stealthy, long-term, accumulative, and irreversible. Some heavy metal elements, such as Cd and Cu, are easy to accumulate in plants, especially food crops, and harm human health through dietary exposure. In this context, remediation of soil contaminated by heavy metals becomes necessary (*Wageh, Hitoshi & Waleed, 2019*; *Silvia, Olga & Alfredo, 2018*). Among the various soil remediation methods, bioremediation is one of the most important (*Selosse, Baudoin & Vandenkoornhuyse, 2004*).

Mycorrhizal fungi can not only promote plant growth but also protect the plant against heavy metal stress (*Wu et al., 2020*; *Chen et al., 2019*; *Meier et al., 2011*; *Moora et al., 2011*; *Gadd, 2007*). Many species of fungi inhabit the forest ecosystems and can resist heavy metal stress (*Li et al., 2020*; *Li et al., 2012*; *Mandyam & Jumpponen, 2005*). However, their mechanisms of resistance to heavy metal stress are unclear. Thus, the ability of mycorrhizal fungi to withstand heavy metal stress needs to be investigated extensively.

Ectomycorrhizal fungi (ECMF) are an important group of fungi, which are ubiquitous in many forest ecosystems (*Yin et al., 2017*; *Yin et al., 2018*). These fungi can increase the nutrient levels in the soil to the levels required by plants for their growth and also help encounter adverse conditions to survive the environmental stress (*Mucha et al., 2006*; *Edda et al., 2010*; *Sharma, Rajak & Pandey, 2010*). ECMF coexist with plant roots, promote plant growth, and even affect the soil microenvironment. Thus, these fungi play an important role in the relationship between plants and the environment (*Regvar et al., 2010*). *Ma (2013)* studied the effect of ECMF on the ability of *Populus × canescens* to absorb and tolerate Cd, and found that ECMF inoculation could improve the resistance of *P. × canescens* to Cd and Cu, so that the plants could grow better.

However, the mechanism by which ECMF resist heavy metal stress is not clear. Relevant studies have shown that ECMF can promote plant growth and enhance plant resistance to heavy metals so that the host plant can grow better under severe stress (*Zhan, Li & Jiang, 2019*; *Zhang, Hu & Yan, 2019*; *Andrade-Linares et al., 2011*; *Abbott, Robson & Boer, 1984*).

Heavy metal stress can induce the production of a large number of reactive oxygen species (ROS), which can induce biological damage. Fungi can adopt various mechanisms to resist the toxicity of heavy metals. It is an important intracellular defense mechanism to secrete antioxidant enzymes to reduce the physiological toxicity caused by heavy metals (*Yan et al., 2017*). For instance, ascorbate peroxidase (APX), catalase (CAT), peroxidase (POD), and superoxide dismutase (SOD) are involved in reducing heavy metal stress and resisting ROS. POD, SOD, and CAT are important ROS-scavenging systems. SOD is the first barrier for cells to resist the stress caused by ROS, which can then be converted into $H_2O_2$, which has a relatively weak oxidation effect. Then, $H_2O_2$ can be decomposed into $H_2O$ by POD and CAT (*Li et al., 2015*). There are many studies on the physiological responses of host plants to heavy metals. However, little is known about the role of antioxidant enzymes, such as ascorbate peroxidase (APX), catalase (CAT), peroxidase (POD), and superoxide dismutase (SOD), in reducing heavy metal stress and resisting reactive oxygen species (ROS) (*Grataõ et al., 2005*; *Hou et al., 2007*; *Zhang et al., 2007*).

Osmotic regulatory substances, including soluble sugar, soluble protein, free proline, and malondialdehyde (MDA), are the products of membrane lipid peroxidation, which

can develop the potential of the lining cells and osmotic pressure in the cells after exposure to environmental stress (*Yin et al., 2018*). Therefore, the changes occurring in the osmotic regulatory substances in the ECM fungal cells under heavy metal stress can be used to evaluate the level of environmental stress (*Yin et al., 2020*).

At present, there is still a lack of basic research data on the development of mycorrhizal agents suitable for heavy metal-contaminated soil, with quantitative research on the stress resistance of Ectomycorrhizal bacteria reported rarely. The present study involved, from the perspective of *Lepista Sordida*'s growth and physiological response, the simulation of the response of two Ectomycorrhizal fungi under heavy metal ($Cd^{2+}$ and $Cu^{2+}$) stress, to provide a theoretical basis for the development of mycorrhizal agents for the remediation of soil polluted with heavy metals.

This study aimed to quantify the growth and physiological responses of ECMF to different Cd and Cu concentrations through the analysis of the antioxidant enzyme activities (APX, CAT, POD, and SOD) and osmotic adjustment substances (soluble sugar, soluble protein, MDA, and free proline). The results of this study can provide a basis for the bioremediation of heavy metal-contaminated soil and the utilization of Ectomycorrhizal fungal resources.

## MATERIAL AND METHODS

### Organism
The fungal strain HLXM obtained from the Liaoning Poplar Research Institute was used in the present study. The strain was grown for ten days on the PDA medium at a pH of around 6.8, as described by *Brundrett et al. (1996)*.

### Molecular biology verification of the taxonomic status of strain
PCR amplification was performed using the ITS1 and ITS4 primers (ITS1, 5′-TCCGTAGGTGAACCTGCGG-3′;ITS4, 5′-TCCTCCGCTTATTATTGATATGC-3′), synthesized by Shanghai Bio-Chemical Co. LTD. The PCR reaction system was: $10 \times$ PCR buffer 5 μL, dNTPS 5 μL, ITS1 primer 5 μL, ITS4 primer 5 μL, Taq enzyme 0.75 μL, DNA template 2.5 μL, ddH$_2$O 26.75 μL, and total volume 50 μL. The PCR reaction conditions were: pre-denaturation at 94 °C for 5 min, denaturation at 94 °C for 30 s, annealing at 56 °C for 45 s, elongation at 72 °C for 2 min, 30 cycles, and supplementation at 72 °C for 10 min. The non-pure PCR products were directly sequenced, and the sequencing results were Blast analyzed in the DNA database in GenBank to determine their classification status. Meanwhile, the sequence was submitted to obtain the gene login number, and the adjacency method in MEGA5.0 software was used to construct the phylogenetic tree for the ITS region (ITS1+ 5.8s +ITS2) for phylogenetic relationship analysis.

### Effects of Cd and Cu stress on strain growth
The strain was cut with a sterile puncher ($\varphi = 10$ mm) and then cultured on solid PDA medium supplemented with different concentrations of Cd and Cu (0, 0.1, 0.2, 0.3, 0.4, and 0.5 mmolL$^{-1}$). The diameter of the mycelium was measured every five days using the cross-sectional method. The state of mycelium of *L. sordida* was determined by scanning

electron microscopy (SEM). The strain was grown in a liquid medium containing Cd and Cu for one week. The aliquot was then filtered to retain the hyphae. After oven-drying the mycelium at 80 °C for 10 h, the hyphae were weighed to measure the dry weight of *L. sordida*.

## Antioxidant enzyme activities of strain

The CAT and POD enzymes were extracted by mixing 0.1 g of hypha in 10 mL of 50 mmol L$^{-1}$ phosphoric acid buffer solution, pH 7.0, followed by centrifuging the mixture at 10,000 rpm for 20 min at 4 °C. The activities of both CAT and POD in the supernatant were analyzed. The SOD and APX enzymes were extracted using 0.1 g of hypha in 10 mL of 50 mmol L$^{-1}$ phosphoric acid buffer solution, pH 7.8. The extract was centrifuged at 10,000 rpm for 20 min at 4 °C, and the activities of total SOD and APX in the supernatant were analyzed.

Commercial kits obtained from Jiancheng, Nanjing, China were used to measure the activities of antioxidant enzymes according to the respective specifications and calculation formulas. These experiments were repeated three times.

## Determination of the contents of osmotic adjustment substances

The content of soluble protein was determined following the method described by *Christos, Konstantinos & George (2008)*. One hundred mg of Coomassie Brilliant Blue (CBB) G-250 100 mg L$^{-1}$ reagent was added to 50 mL of 95% ethanol and mixed until dissolved. One hundred mL of 15 molL$^{-1}$ H$_3$PO$_4$ was added to 500 mL of distilled H$_2$O and mixed well. Hyphae (0.5 g) were ground in 10 mL distilled water and centrifuged at 10,000 rpm for 5 min. Finally, 1 mL supernatant was added to 5 mL CBB reagent and mixed well. The absorbance was recorded at 595 nm. The experiment was repeated three times.

The content of soluble sugars was analyzed using the anthrone method described by *Yemm & Willis (1954)*. The fine powder of hyphae (about 100 mg) was homogenized in 3 mL of 80% ethanol, followed by incubation in an ultrasonic bath at 80 °C for 30 min. After centrifugation (6000 g, 25 °C, 10 min), the supernatant was collected. The pellet was extracted as above, and the supernatant was collected and combined with the previous one. After adding 2 mL of the anthrone reagent to the supernatant, the mixture was heated in boiling water for 7 min. After the mixture was cooled to room temperature, the absorbance of the mixture was recorded spectrophotometrically at 620 nm. The experiment was repeated three times.

Malondialdehyde (MDA) levels were measured as described by *Wasowicz, Jean & Peratz (1993)*. About 0.2 g of hyphae fresh tissues were placed in a tube containing 1 mL distilled water. After the addition of 1 mL of 29 mmol L$^{-1}$ acetic acid solution, the samples were placed in a water bath and heated for 1 h at 95-−100 °C. After the samples were cooled under running water, 25 μL 5 mol L$^{-1}$ HCl was added, and the reaction was stopped by adding 3.5 mL of n-butanol and agitating for 5 min. After centrifugation (10,000 rpm for 5 min), the separated butanol phase was removed, and the fluorescence was measured in a spectrofluorometer (Shimadzu-RF-5000, Kyoto, Japan) using 525 nm for excitation and 547 nm for emission. The experiment was repeated three times.

Proline in the hyphae tissues was analyzed using the method described by *Bates, Waldren & Teare (1973)*. About 0.5 g of frozen hyphae fresh tissue was ground in liquid $N_2$. The powder was then mixed into 1 mL of aqueous sulfosalicylic acid (3%, w/v). This solution was then mixed with an equal volume of glacial acetic acid and ninhydrin reagent and heated at 95 °C for 1 h. The reaction was terminated by placing the container in an ice bath for 15 min, and the reaction solution was mixed with 2 mL of toluene. The absorbance of the chromophore was measured at 520 nm in a UV–VIS spectrophotometer (HACH DR/4000; model 48000, HACH Co., Loveland, Colorado, USA) and compared with the absorbance of a standard. The experiment was repeated three times.

## Statistical analysis

Data processing was performed using Excel 2010, and one-way analysis of variance (ANOVA) was performed using GraphPad Prism 7.0. Multiple comparison tests for different treatments were conducted using Duncan's multiple range test with the significance levels ($\alpha$) of 0.05, 0.01, and 0.001. The charts were drawn using GraphPad Prism 7.0.

# RESULTS

## Molecular biological verification of the taxonomic status of strain

After determining the ITS sequence of the strain and verifying its ITS classification status, the length of the ITS sequence PCR product of the strain was determined to be 643 bp. MEGA5.0 software adjacency method was used to construct the phylogenetic tree, and the sequence similarity rate between the strain and *Lepista Sordida* was up to 99.69%. The sequence was submitted to GenBank to obtain the gene login number: MT645231 (https://www.ncbi.nlm.nih.gov/nuccore/MT645231) (Fig. 1).

## Morphological observation of *L. sordida* under Cd and Cu stress

The growth of *L. sordida* on the PDA medium supplemented with Cd and Cu was monitored for 25 days. There were considerable changes in the hyphal diameter in the media containing different Cd concentrations compared to the control (0 mmol $L^{-1}$) (Fig. 2A). However, mycelial growth was not significantly inhibited under the effect of heavy metals (Fig. 2B). With increasing Cd concentration, the diameter of the colonies gradually decreased, and the growth also slowed down. At the end of the experiment (25 days), the colonies in the control group had covered the whole Petri dish (Fig. 3A), while those in the Cd stress group (Figs. 3B–3F) had not covered the Petri dishes. When the concentration of Cd increased from 0.1 to 0.5 mmol $L^{-1}$, the diameter of the colony decreased successively, and the growth also slowed down gradually. The mycelial colony gradually became sparse.

The difference is that the increasing Cu concentration did not result in a significant decrease in the colony diameter. At the end of the experiment (25 days), although the colonies in the control group had covered the whole Petri dish (Fig. 4A), those in the Cu stress group (Figs. 4B–4F) had not covered the Petri dishes. However, when the concentration of Cu increased from 0.1 to 0.5 mmol $L^{-1}$, the decrease in the colony

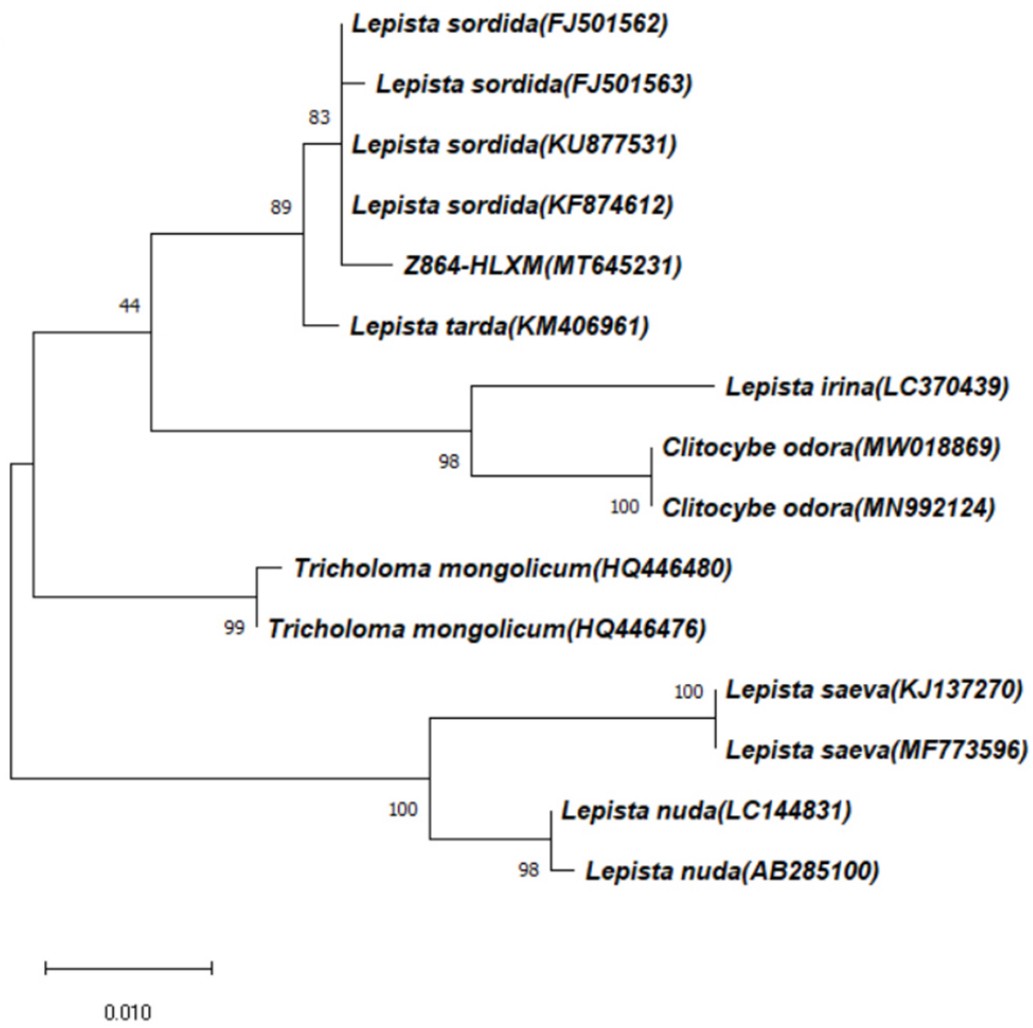

**Figure 1  Genetic evolutionary tree of strain.**

diameter was not significant. The mycelial colony gradually became sparse. The results showed that *L. sordida* could successfully resist the stress caused by heavy metals.

The morphological changes in the mycelium of *L. sordida* were observed by SEM. Cadmium (Cd) caused conspicuous changes in the hyphal morphology (Fig. 5). The control group presented profound and luxuriant mycelium growth, with numerous clamp connections (Fig. 5A). A bright contrast was observed between the control group (Figs. 5B–5F) and the Cd stress group, with the observation of twisting of individual hyphae under the influence of Cd (Fig. 5C) and the breaking of certain mycelia (indicated by the arrow) (Figs. 5D–5E). When the concentration of Cd was 0.5 mmolL$^{-1}$, the mycelium started dissolving, as indicated by the arrow (Fig. 5F).

Copper (Cu), at particularly high concentrations, caused conspicuous changes in the hyphal morphology (Fig. 6F). When the concentration increased from 0 to 0.2 mmolL$^{-1}$ in the Cu stress groups, the mycelium growth was profound and luxuriant, with numerous

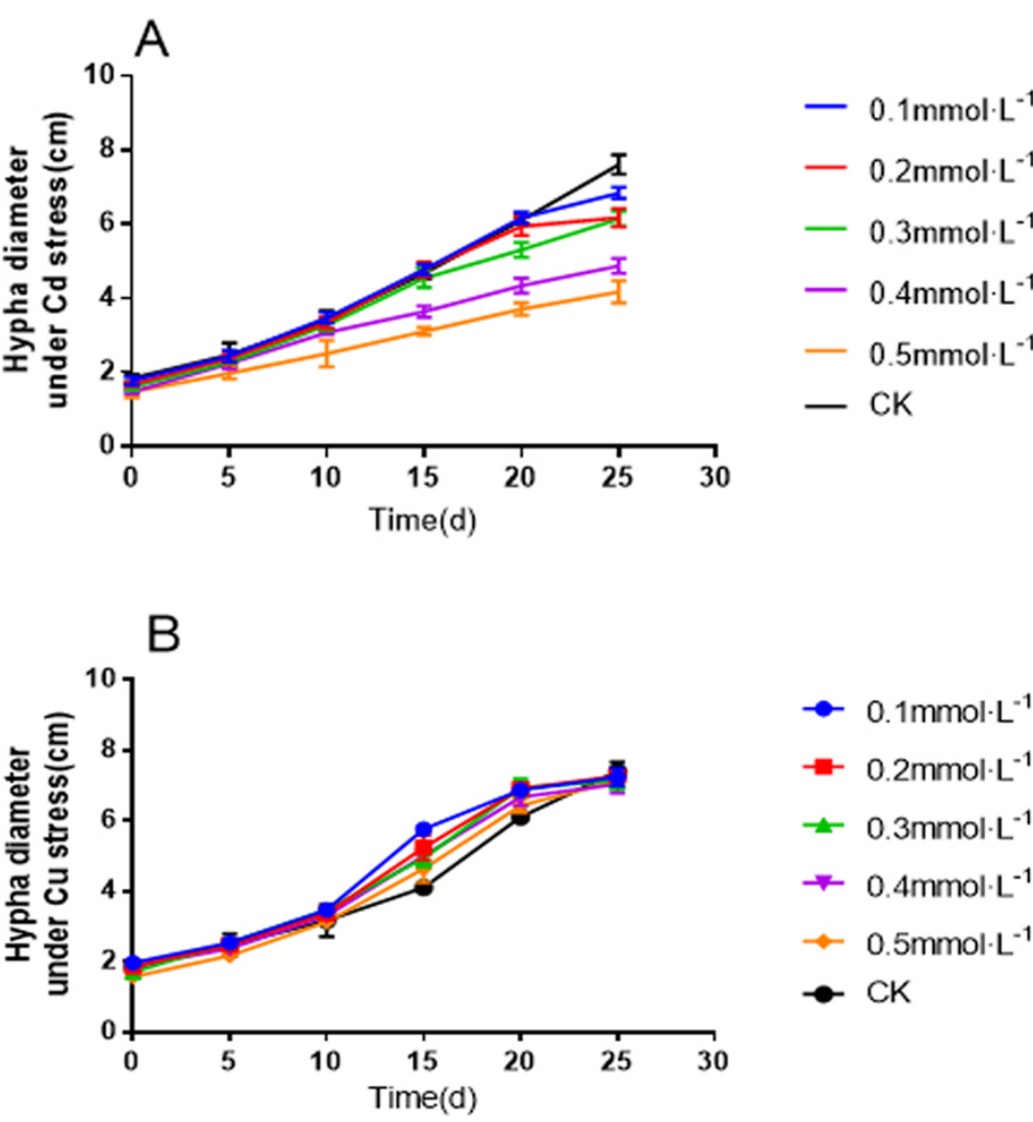

**Figure 2** Twenty-five-day-old hyphal diameter of *L. sordida* on a PDA medium: (A) Mycelial growth curve under Cd stress; (B) Mycelial growth curve under Cd stress.

clamp connections, as indicated by the arrow (Figs. 6A–6C). There was a high contrast between the control group and the Cu stress group, with a twisting of the intertwined hyphal strands under the effect of Cu (indicated by the arrow) (Figs. 6D–6F). When the concentration of Cu was 0.5 mmol $L^{-1}$, the mycelium started breaking and dissolving, as indicated by the arrow (Fig. 6F).

The growth of *L. sordida* in the liquid medium containing Cd was observed. There were remarkable differences in the mycelial dry weight under Cd and Cu stress. At higher concentrations of Cd and Cu (0.4–0.5 mmol $L^{-1}$), the mycelial dry weight was significantly different from that in the control group ($P < 0.05$ and $P < 0.01$, respectively) (Fig. 7A). The results showed that *L. sordida* could tolerate low concentrations of heavy metals.

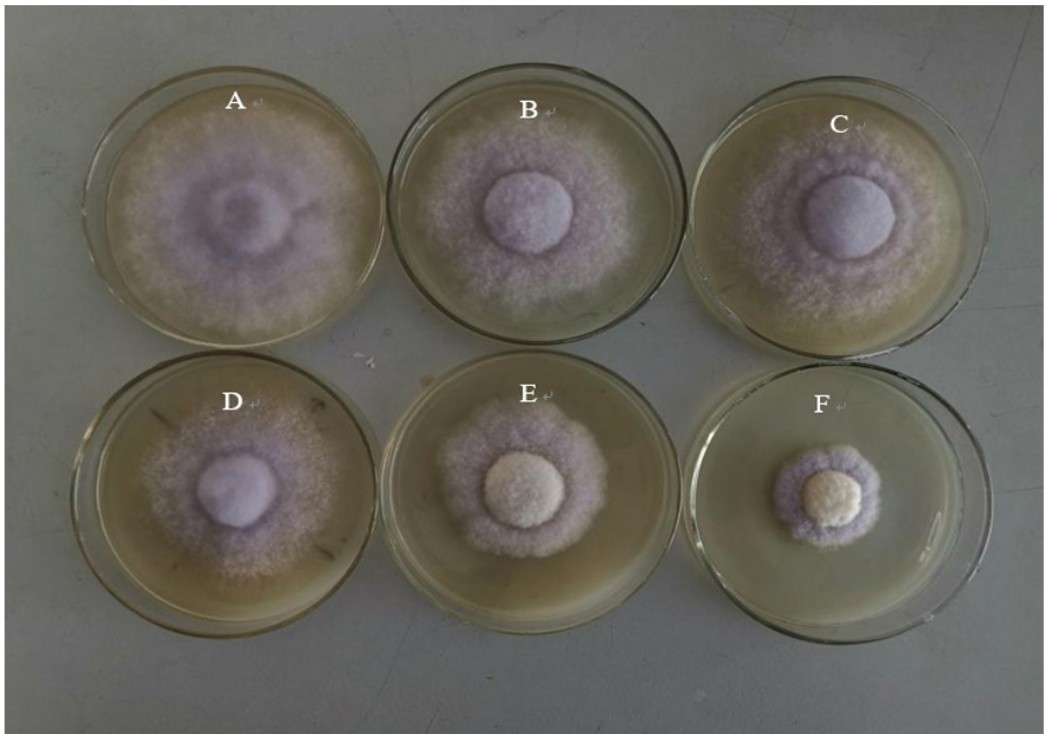

**Figure 3** **The growth of *L. sordida* exposed to different concentrations of Cd for 25 days.** (A) The growth of *L. sordida* with no Cd stress; (B) The growth of *L. sordida* on PDA medium with 0.1 mmol L$^{-1}$ of Cd; (C) The growth of *L. sordida* on PDA medium with 0.2 mmol L$^{-1}$ of Cd; (D) The growth of *L. sordida* on PDA medium with 0.3 mmol L$^{-1}$ of Cd; (E) The growth of *L. sordida* on PDA medium with 0.4 mmol L$^{-1}$ of Cd; (F) The growth of *L. sordida* on PDA medium with 0.5 mmol L$^{-1}$ of Cd.

## Antioxidant enzyme activities of *L. sordida* under Cd and Cu stress

As shown in Fig. 8, When the Cd concentrations were 0.2 and 0.3 mmolL$^{-1}$, the activities of CAT in *L. sordida* were 66% and 233% higher than those in the control, with significant differences between them ($P < 0.05$ and $P < 0.0001$, respectively) (Fig. 8A). Similarly, when the Cu concentrations were 0.3 and 0.4 mmol L$^{-1}$, the activities of CAT in *L. sordida* were 100% and 56% higher than those in the control, with significant differences between the groups ($P < 0.001$ and $P < 0.05$, respectively) (Fig. 8B). The effect of Cd on the activity of POD in *L. sordida* was more significant than that on its CAT activity (Fig. 8C). Furthermore, the POD activity in *L. sordida* exposed to 0.4 mmol L$^{-1}$ was significantly different ($P < 0.05$) from that of the control (Fig. 8D). One explanation for the inconsistently varied pattern of POD under Cd and Cu stress could be that Cu$^{2+}$ is a trace element for this organism and, therefore, a low concentration of Cu$^{2+}$ would not exert a great impact on the organism, while a higher concentration of Cu$^{2+}$ would exert a greater impact on the growth and physiological metabolism of the organism. Nonetheless, the specific reasons for this require further investigation.

As shown in Fig. 9, when the Cd concentrations were 0.2 and 0.3 mmol L$^{-1}$, the activities of SOD in *L. sordida* were 100% and 45.5% higher than those in the control, respectively.

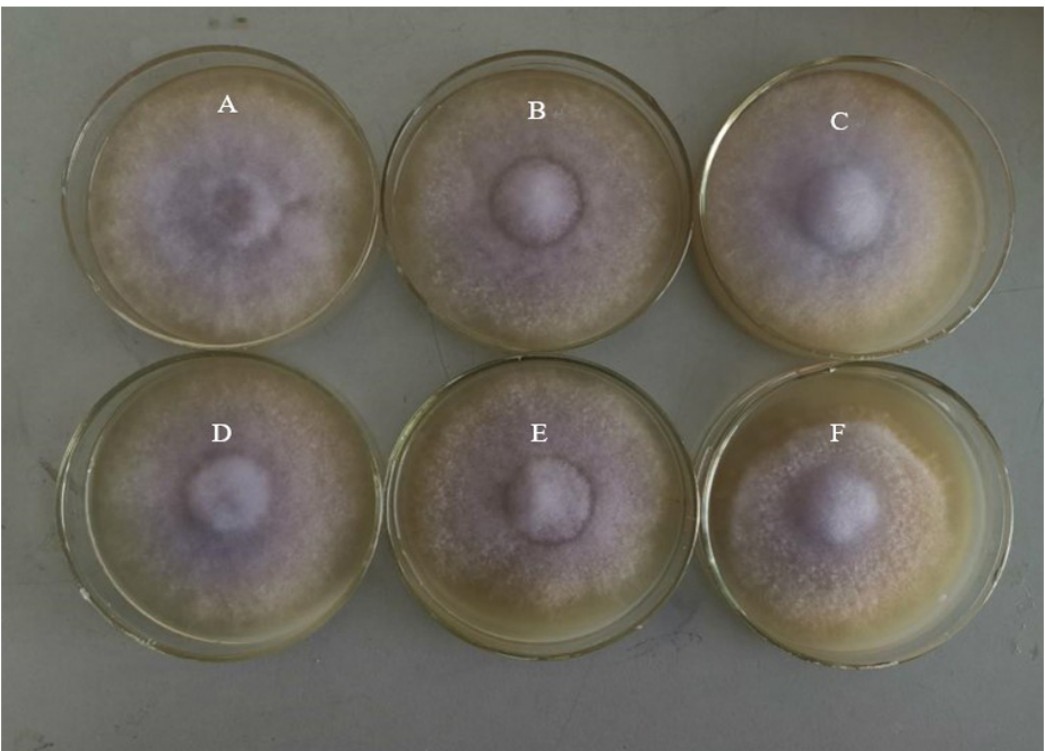

**Figure 4 The growth of *L. sordida* exposed to different concentrations of Cu for 25 days.** (A) The growth of *L. sordida* with no Cu stress; (B) The growth of *L. sordida* on PDA medium with 0.1 mmol L$^{-1}$ of Cu; (C) The growth of *L. sordida* on PDA medium with 0.2 mmol L$^{-1}$ of Cu; (D) The growth of *L. sordida* on PDA medium with 0.3 mmol L$^{-1}$ of Cu; (E) The growth of *L. sordida* on PDA medium with 0.4 mmol L$^{-1}$ of Cu; (F) The growth of *L. sordida* on PDA medium with 0.5 mmol L$^{-1}$ of Cu.

Similarly, when the Cu concentrations were 0.2 and 0.3 mmol L$^{-1}$, the activities of SOD were 82% and 91% higher than those in the control, with significant differences ($P < 0.05$ and $P < 0.001$), respectively, between the treatments (Fig. 9B). The effect of Cd on the activity of APX in *L. sordida* was more significant than that on its SOD activity (Fig. 9C). Moreover, the POD activity in *L. sordida* exposed to 0.2 mmol L$^{-1}$ was significantly different ($P < 0.01$) from that of the control (Fig. 9D).

## The contents of osmotic adjustment substances in *L. sordida* under Cd and Cu stress

The contents of soluble sugar and protein in *L. sordida* under Cd and Cu stress presented in Fig. 10 indicate that the addition of Cd and Cu could affect the soluble sugar and protein contents in *L. sordida*. Higher Cd concentrations, particularly 0.1 and 0.2 mmolL$^{-1}$, increased the soluble sugar and protein contents (Fig. 10A). Similarly, when the concentration of Cu was between 0.1 and 0.3 mmol L$^{-1}$, the contents of soluble sugar in *L. sordida* were 71%, 57%, and 86% higher, respectively, than those in the control, with significant differences between the groups ($P < 0.01$, $P < 0.05$, $P < 0.01$) (Fig. 10B). The

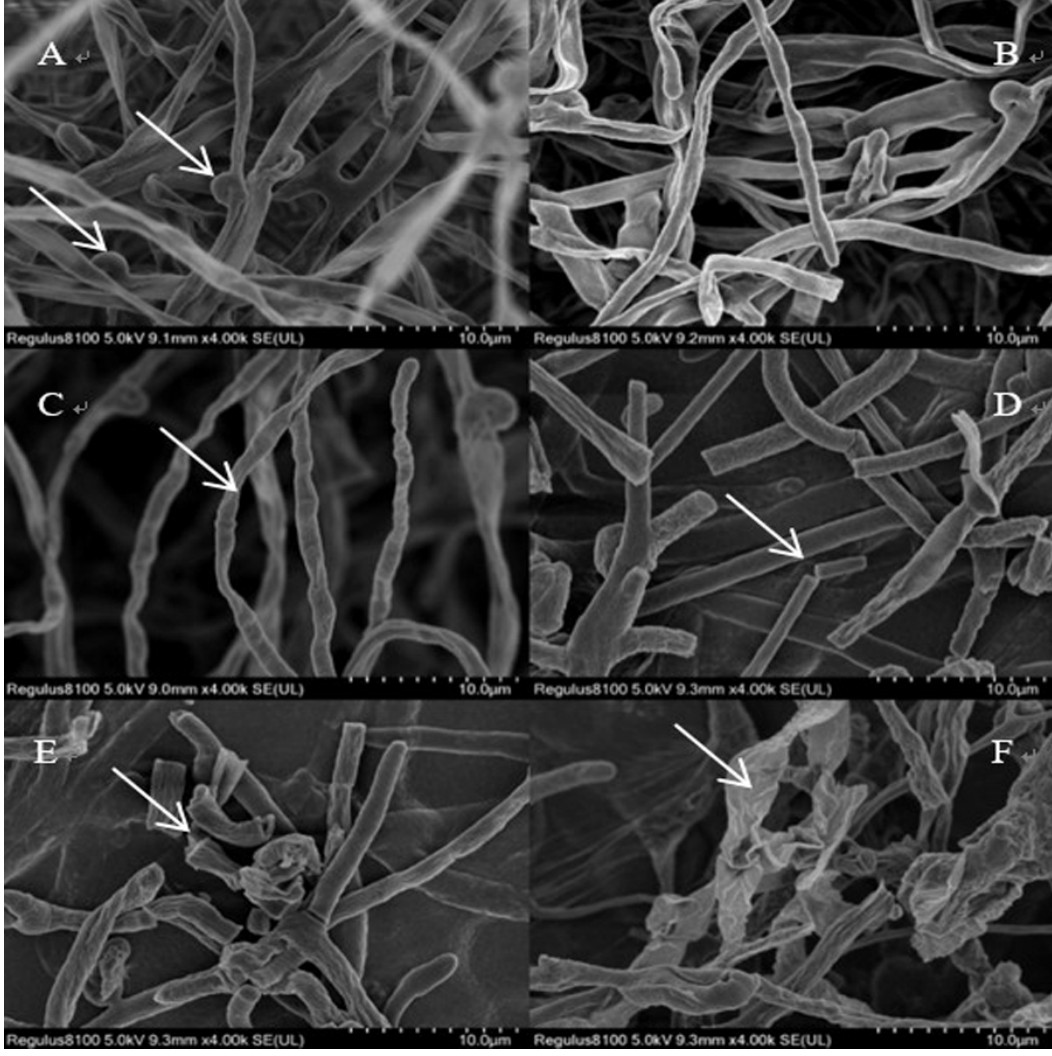

**Figure 5** **Growing status of *L. sordida* on PDA medium under Cd stress.** SEM of the hyphae from the colony edge of *L. sordida*. After adding Cd to the medium, the mycelium will twist and break, stick to each other and even dissolve. Control group mycelial growth is good, the mycelium is luxuriant with full of clamp connections. (A) Hyphae from the 0 mmol L$^{-1}$; (B) Hyphae from a colony treated with 0.1 mmol L$^{-1}$ of Cd; (C-F) Mycelial special morphology of *L. sordida* under Cd stress from 0.2 mmol L$^{-1}$ to 0.5 mmol L$^{-1}$.

effect of Cu on the content of soluble protein in *L. sordida* was more significant than on the control (Fig. 10D).

Figure 11 demonstrates the changes in the *malondialdehyde* (MDA) levels and free proline content in *L. sordida* as a result of the addition of Cd and Cu to the medium. Higher Cd concentrations, particularly 0.3–0.5 mmol L$^{-1}$, increased the MDA content significantly (with significance differences of $P < 0.01$, $P < 0.0001$, and $P < 0.0001$, respectively) (Fig. 11A). Similarly, when the Cu concentration ranged from 0.3 to 0.5 mmol L$^{-1}$, the contents of soluble sugar in *L. sordida* were 50%, 150%, and 225% higher,

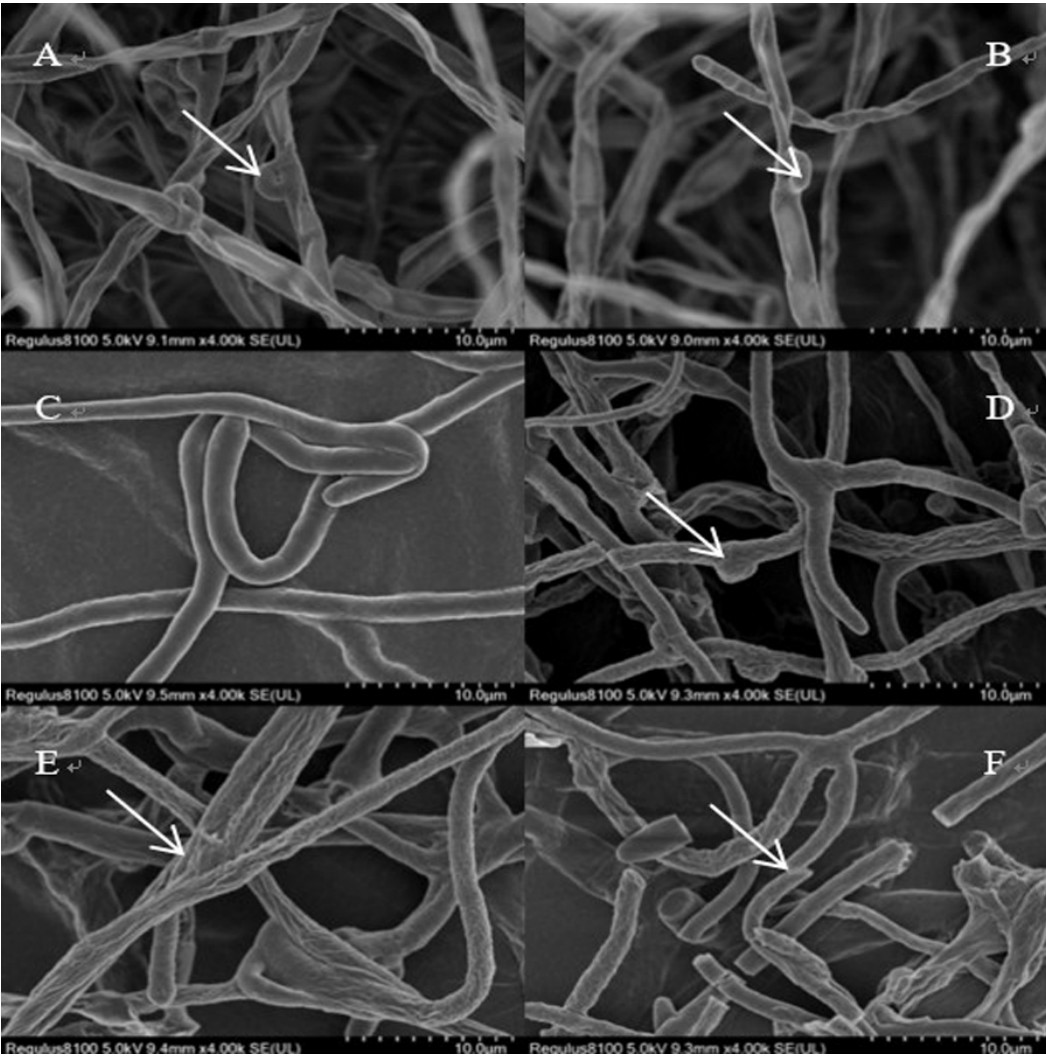

**Figure 6  Growing status of *L. sordida* on PDA medium under Cu stress.**  SEM of the hyphae from the colony edge of *L. sordida*. After adding Cu to the medium, the mycelium will twist and break, sunken. Control group mycelium growth is good, mycelium is luxuriant with full of clamp connections. (A) Hyphae from the 0 mmol $L^{-1}$; (B) Hyphae from a colony treated with 0.1 mmol $L^{-1}$ of Cu; (C–F) Mycelial special morphology of *L. sordida* under Cu stress from 0.2 mmol $L^{-1}$ to 0.5 mmol $L^{-1}$.

respectively, than those in the control, with significant differences ($P < 0.05$, $P < 0.0001$, $P < 0.0001$) between the groups (Fig. 11B). The effects of Cd and Cu on the content of free proline in *L. sordida* were more significant compared to those on the control (Figs. 11C and 11D).

# DISCUSSION

Soil remediation using a combination of plants and microorganisms has become a research hotspot nowadays (*Chaney, 1983*; *Jamal, Ayub & Usman, 2002*; *Hu, Lin & Wang, 2009*; *Jack, Jose & Joanne, 2010*). In recent years, scholars have attempted to use woody plants

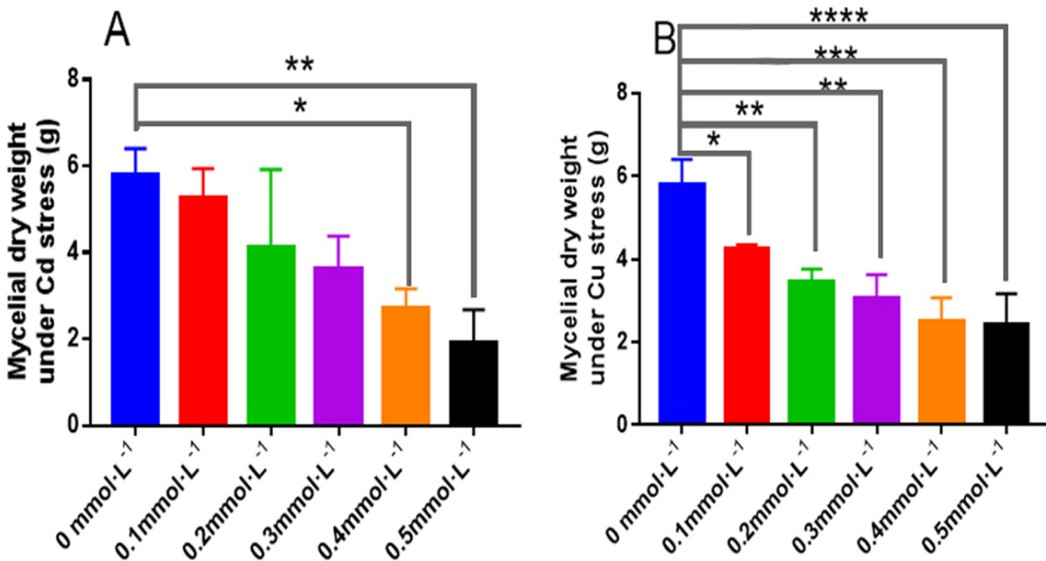

**Figure 7** **The biomass of L. sordida treated with Cd and Cu for 10 days.** (A) The biomass of L. sordida under Cd stress; (B) the biomass of L. sordida under Cu stress.

with more abundant biomass for restoration and exploration. *Zhang, Chai & Wang (2013)* used the mycorrhizal fungi *Pinus tabulaeformis* for the remediation of soil contaminated with Cd and suggested that inoculation with heat-resistant proteins secreted by exogenous mycorrhizal fungi could significantly improve the fixation capacity of heavy metals in the rhizosphere of *P. tabulaeformis*. It was found that ECM could protect the aboveground parts of birch seedlings under high concentrations of Cu and Cd. It is noteworthy that understanding the mechanisms of heavy metal tolerance in mycorrhizal fungi is important for improving the soil using mycorrhizal fungi.

In this study, Cd and Cu stress were observed to significantly affect the growth of mycorrhizal fungi. After the addition of Cd and Cu to the medium, mycelium started twisting, breaking, and finally dissolved. The mycelial growth in the control group was profound, luxuriant, and with numerous clamp connections. Moreover, the mycorrhizal fungi had better growth at low concentrations of heavy metals. The results showed that ECMF had a certain degree of tolerance to heavy metals and could grow normally in the presence of heavy metals.

When the concentration of Cu increased from 0.1 to 0.5 mmol $L^{-1}$, the decrease in the colony diameter was not significant (Fig. 4). However the mycelial dry weight under Cd stress was significantly different from that in the control group in Fig. 7B. The reason may be that the toxicity of the two heavy metal ions is different. Cd is a highly toxic heavy metal, while Cu is a micronutrient element of organisms, and when the concentration is too high, it will cause adverse effects on the growth of organisms, or even toxic effects. However, Cd is different. Cd itself is a highly toxic heavy metal element. Under the regulation of Cd, it will cause great harm to organisms.

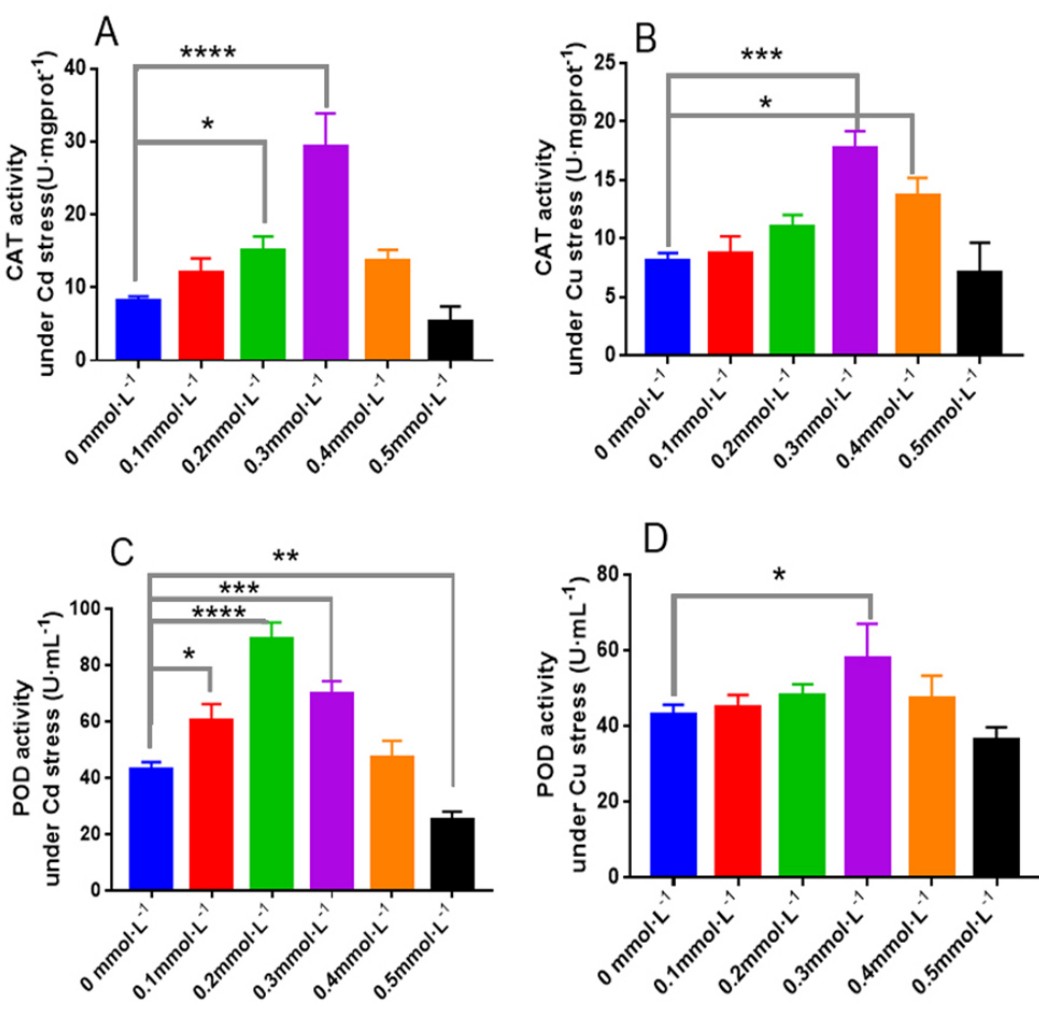

**Figure 8** **The responses of CAT and POD to the toxicity of Cd and Cu for 25 days.** (A) The activity of CAT under Cd stress; (B) the activity of CAT under Cd stress; (C) the activity of POD under Cu stress; (D) the activity of POD under Cu stress. * indicate significant differences ($P < 0.05$) assessed by Duncan's test. ** was $P < 0.01$; *** was $P < 0.001$; **** was $P < 0.0001$. Data are means ± SE ($n = 3$).

CAT, POD, SOD, and APX are the protective enzymes occurring widely in animals, plants, and microorganisms (*Yan et al., 2017*). When in coordination with each other, they can maintain the dynamic balance between the production and removal of oxygen free radicals in the cells, remove ROS produced due to environmental stress, reduce the oxidative stress caused by ROS in cells, and thus prevent the toxicity of oxygen free radicals (*Halliwell & Gutteridge, 2007*). The activities of antioxidant enzymes in *L. sordida* were analyzed to study the responses of anti-reactive oxygen capacity to Cd and Cu stress. The toxic effects of Cd and Cu appeared to be related to the production of ROS and could cause cell rupture. Therefore, the degree of destruction of the ectomycorrhizal cell under Cd and Cu stress can be represented by the changes in the antioxidant enzyme activities and osmotic regulatory substances (*Baldrian, 2003*; *Bai, Harvey & McNeil, 2003*; *Halliwell*

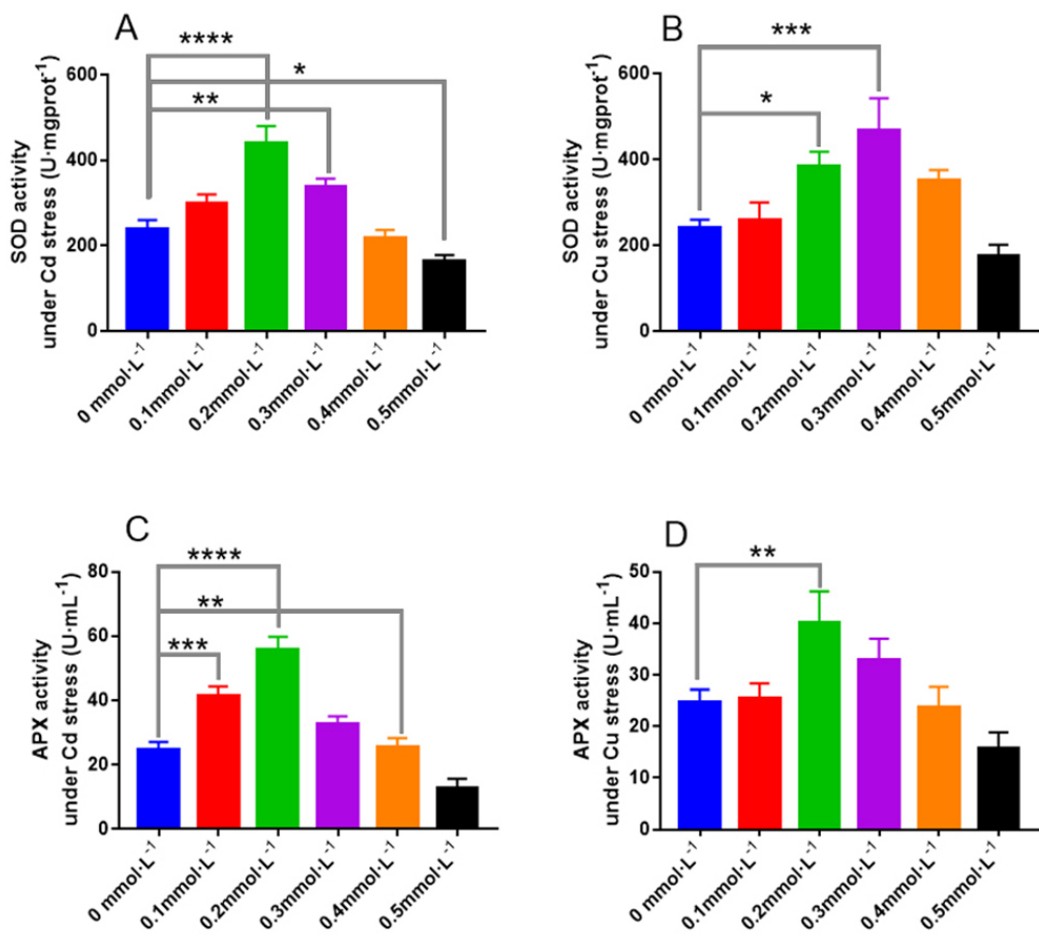

**Figure 9** **The responses of SOD and APX to the toxicity of Cd and Cu for 25 days.** (A) The activity of SOD under Cd stress; (B) the activity of SOD under Cd stress; (C) the activity of APX under Cu stress; (D) the activity of APX under Cu stress. * indicate significant differences ($P < 0.05$) assessed by Duncan's test. ** was $P < 0.01$; ***was $P < 0.001$; **** was $P < 0.0001$. Data are means ± SE ($n = 3$).

*& Gutteridge, 2007*). In this study, the antioxidant enzyme activities in *L. sordida* increased when exposed to 0.2 and 0.3 mmol $L^{-1}$ Cu and Cd. However, upon exposure to 0.4 and 0.5 mmol $L^{-1}$ of these metals, the antioxidant enzyme activities decreased. APX, POD, and SOD had the highest activities at the $Cd^{2+}$ and$Cu^{2+}$ concentrations of was 0.2 mmol $L^{-1}$. CAT activity reached its maximum value at a metal concentration of 0.3 mmol $L^{-1}$, indicating that the reaction time of CAT for ROS induction was longer than that of the other three enzymes. The reason might be the effect of high concentrations of Cd and Cu, which resulted in cell death in *L. sordida*. The results showed that the fungus *L. sordida* can grow normally under heavy metal stress, has a certain degree of tolerance to heavy metals, can resist the ROS, and remove oxygen radicals and their products via the production of antioxidant enzymes and increasing their activities (*Halliwell & Gutteridge, 2007*). These results were similar to those reported by *Hegedu et al. (2007)*.

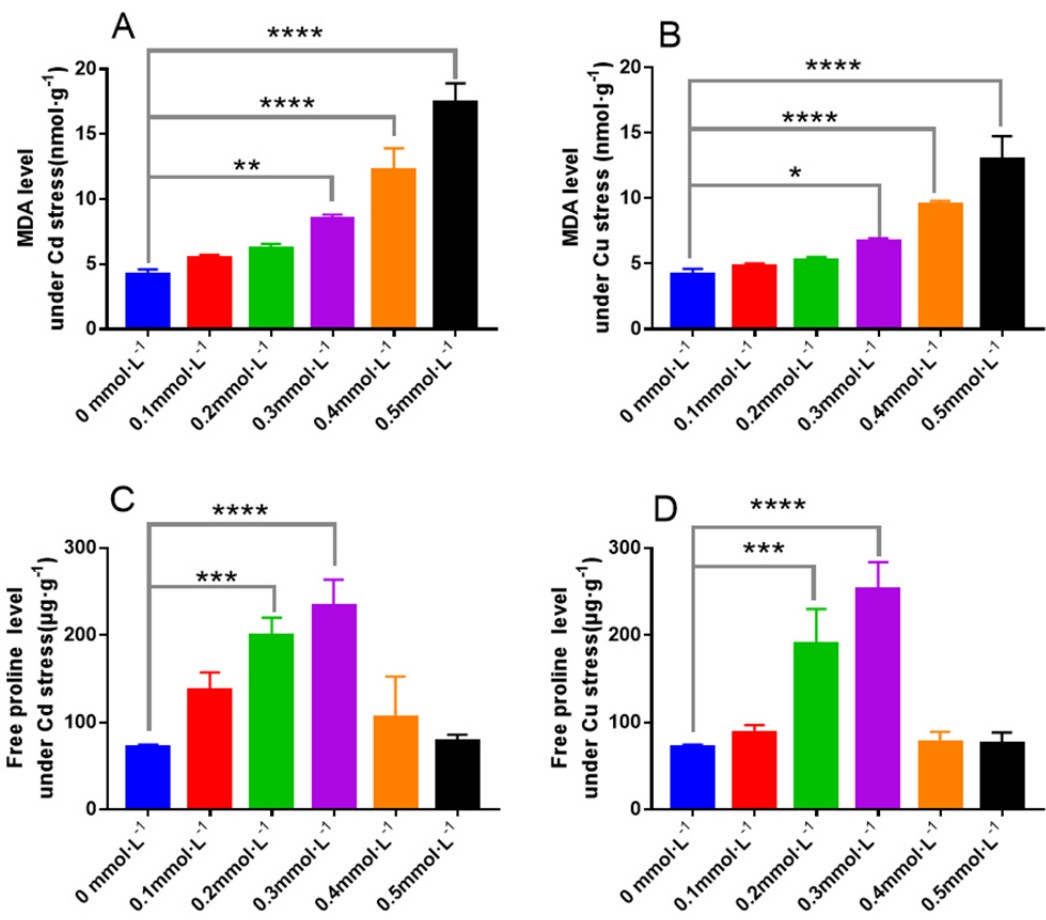

**Figure 11** **The responses of MDA and free proline to the toxicity of Cd and Cu for 25 days.** (A) The content of MDA under Cd stress; (B) the content of MDA under Cd stress; (C) the content of free proline under Cu stress; (D) the content of free proline under Cu stress. * indicate significant differences ($P <$ 0.05) assessed by Duncan's test. ** was $P < 0.01$; *** was $P < 0.001$; **** was $P < 0.0001$. Data are means $\pm$ SE ($n = 3$).

from those in the control group, indicating that the low concentrations of Cd and Cu had no significant effects on the Ectomycorrhizal cells.

The content of MDA increased with the increase of heavy metal concentration, indicating the aggravation of heavy metal ion damage to *L. sordida* cells. These results corresponded with the previous increase in soluble protein and soluble sugar content to varying degrees, indicating that heavy metal ions had a certain destructive effect on *L. sordida* cells. However, under low concentration of heavy metals stress, the MDA content of *L. sordida* cells was not significantly affected, indicating that *L. sordida* had a certain tolerance to heavy metals.

The increase in the free proline content in organisms is a physiological and biochemical response to stress. In this study, free proline content in the fungus was higher in Cd and Cu treatment groups compared to the control group, regardless of Cd and Cu stress, and it followed a pattern similar to that of the antioxidant enzyme activities. When the concentrations of heavy metals were 0.1 and 0.2 mmol L$^{-1}$, the free proline content in

ECMF was not significantly different ($P > 0.05$) from that of the control group, indicating that low concentrations of Cd and Cu had no significant effect on the cells of ECMF.

In conclusion, the results of the present study can provide a theoretical basis for the better utilization of ECM fungal resources for the remediation of soil contaminated with heavy metals.

## CONCLUSION

*Lepista sordida*, an Ectomycorrhizal (ECM) fungus, can resist against Cd and Cu. The fungus *L. sordida* was observed for its growth, antioxidant enzyme activities, and osmotic regulation. These indicators can reflect strong resistance to heavy metal stress in *L. sordida*. Besides, this study provides necessary data for further investigations aiming for better utilization of Ectomycorrhizal fungal resources for the remediation of heavy metal-contaminated soil.

### Funding
The research was supported by the National Natural Science Foundation of China (31800542). The funders had no role in study design, data collection and analysis, decision to publish, or preparation of the manuscript.

### Grant Disclosures
The following grant information was disclosed by the authors:
National Natural Science Foundation of China: 31800542.

### Competing Interests
The authors declare there are no competing interests.

### Author Contributions
- Yin Dachuan conceived and designed the experiments, performed the experiments, prepared figures and/or tables, and approved the final draft.
- Qi Jinyu analyzed the data, authored or reviewed drafts of the paper, and approved the final draft.

### DNA Deposition
The following information was supplied regarding the deposition of DNA sequences:
The sequence is available at GenBank: https://www.ncbi.nlm.nih.gov/nuccore/MT645231.

### Data Availability
Raw measurements are available in the Supplemental Files.

## Supplemental Information

Supplemental information for this article can be found online at http://dx.doi.org/10.7717/peerj.11115#supplemental-information.

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
