# Peer review of "The physiological response of Ectomycorrhizal fungus Lepista sordida to Cd and Cu stress"

_PeerJ, doi:10.7717/peerj.11115_

## Round 0.1 · original submission · Major Revisions

This paper can be considered after substantial revisions.

·

Basic reporting

Please see my general comments for improvements.

Experimental design

Please see my general comments for improvements.

Validity of the findings

No comment.

Additional comments

I have reviewed this manuscript for its suitability for PeerJ.

The authors aimed to investigate the response of the ectomycorrhizal fungus Lepista sordida to Cd and Cu stress in pure culture.

Studies regarding growth and physiological indexes of L. sordida under Cd and Cu stress were used. However, the article is premature and needs significant additional input from the authors to be published in a journal like PeerJ, especially relative to the lack of experiments carried out to investigate the effects of this fungi (AMF) on the remediation potential in Cd or Cu-polluted soils. Furthermore, the authors should consider adding the quantification of non-enzymatic molecules with antioxidant function (and/or the activity of glutathione-related antioxidant enzymes), besides quantifying reactive oxygen species (such as hydrogen peroxide) to better characterize the antioxidant response and oxidative stress.


For these main reasons, I feel the submitted study now is not of sufficient quality to be published in this journal. Additionally, I provide the following comments for improving the manuscript:



1) In lines 30 to 32, heavy metal contamination was introduced. However, there are other heavy metals besides Cd and Cu. Therefore, I suggest the authors explain why they selected Cd and Cu for the present study. Furthermore, the authors should introduce and explain the importance of studying the toxicity of Cd and Cu stress.

2) Regarding the sentences ´´Mycorrhizal fungi not only can promote plant growth but also protect it against heavy metal stress`` (lines 35 and 36) and ´´Many species of fungi inhabit forest 37 ecosystems, and many can resist heavy metal stress` `(lines 36 and 37), I suggest the authors also provide citations of recent articles.

3) Regarding ´´Relevant 50 studies have shown that ECMF can promote plant growth and enhance plant resistance to heavy 51 metals so that the host plant can grow better under severe stress`` (lines 49 to 51), I suggest the authors cite recent articles.

4) In the introduction, the authors should provide more examples of studies focused on Cd and Cu stress (instead of addressing heavy metal in general), citing recent articles.


5) Instead of writing about heavy metals in general, I suggest the authors rewrite the sentence: ´´ However, little is known about the role of antioxidant enzymes such as ascorbate peroxidase 55 (APX), catalase (CAT), peroxidase (POD) and superoxide dismutase (SOD) in reducing heavy 56 metal stress and resist reactive oxygen species (ROS) `` (lines 54 to 56), citing recent studies that evaluated the enzymatic antioxidant response to Cd and Cu stress. Furthermore, in addition to not being included in the list of references, the citations ´´ Grata˜ et al. 2005; Hou et al. 2007; 57 Zhang et al. 2007) `` are too old to write that ´´..little is known about the role of antioxidant enzymes ... ''.

6) For the introduction, I suggest that the authors explain better on how this study can provide a basis for the bioremediation of heavy metal contaminated soil.


7) Regarding introduction, the authors should provide the context, explain and better introduce the role of antioxidant enzymes related to this study. Since there are several molecules involved in modulating the Cd and Cu stress response besides antioxidante molecules, I suggest the authors to explain why antioxidant enzymes deserve special attention for the present research.

8) For the introduction, I suggest the authors introduce oxidative stress induced by heavy metals (especially by Cd and Cu stress) and introduce reactive oxygen species


9) Regarding introduction and discussion, the authors should explain the difference of the role of the antioxidant enzymes assessed in this study regarding their role in scavenging of reactive oxygen species citing articles.

10) I suggest the authors to provide scientific citation for the following sentence (or rewrite it):´´ Osmotic regulatory substances, including soluble sugar, soluble protein, free proline, and
malondialdehyde (MDA) are membrane lipid peroxidation products, which can develop the potential of lining cells and osmotic pressure in the cells after exposure to environmental stress`` (lines 58 to 60).


11) Regarding methods, I suggest the authors provide information on how the activity of the antioxidant enzymes was calculated.

12) Regarding the following sentence, I suggest the authors replace the word ´´provoke`` with ´´study`` (or another one to improve the meaning of the sentence): ´´The activities of antioxidant enzymes in L. sordida were analyzed to provoke the response of anti-reactive oxygen capacity to Cd and Cu stress`` (lines 241 and 242).

13) The discussion is too short. It should include more information including the role of antioxidant enzymes in the cell.


14) The authors should be careful when it comes to figure captions. In figures 8, 9, 10 and 11, ´´under Cd stress`` was written for both ´´A`` and ´´B``. In Figure 7, the authors should provide information regarding ´´A`` and ´´B``.


15) The authors should not italicize ´´an ectomycorrhiza`` (line 276) and ´´was observed `` (line 277).

Reviewer 2 ·

Basic reporting

This paper showed that the growth and physiological changes of Lepista sordida under different Cd and Cu concentrations in pure culture. By SEM, authors found that the mycelium twisted, broke, sticked together, and even dissolved under Cd and Cu stress. Also, authors indicated the response of Lepista sordida to the different Cd and Cu concentrations by analyzing antioxidant enzyme activities and osmotic adjustment substances. However, the study did not explain whether ECMF has metal tolerance, and why. Finally, the study only described some results without forming a confirm conclusion

Experimental design

Authors need to provide more species to build the more comprehensive evolutionary tree

Validity of the findings

1. The introduction parts needs more information, such as why chose Zn and Cd as HM stress for this fungus.
2. Add more references to support the explanation of experimental results. The explanation for the inconsistency of the varied pattern of POD under Cd and Cu stress? Cu as a trace element?

Additional comments

The detailed comments are as follows.


1. The English language should be improved to ensure that the international audience can clearly understand your paper.
2. L74-75, ITS primers, wording style.
3. L75-76, Is it L or μL?
4. L115, 60009g? Is the statement correct?

---

## Round 0.2 · Major Revisions

Dear Yin,
I have received the review report. Although a reviewer is positive the other reviewer asked for some additional data and drastic revisions.
Therefore, it can not be considered for publication in its current state but I encourage a revision if you can solve all the issues.

Reviewer 2 ·

Basic reporting

Some format problems may be due to format conversion please check , for example Fig 3,4,5 and 6 explanatory text

Experimental design

no comment

Validity of the findings

no comment

Additional comments

It is recommended that the error line of the histogram should not be the same color as the column.

Reviewer 3 ·

Basic reporting

No comment.

Experimental design

No comment.

Validity of the findings

No comment.

Additional comments

Reviewer’s comments:
The manuscript entitled "The Response of Ectomycorrhizal fungus Lepista sordida to Cd and Cu stress in Pure Culture (#55703)" has been submitted by Yin and Qi. This study aimed to investigate the effects of Cd and Cu exposure on growth and physiological indices of EMF Lepista sordida. However, in my opinion the work lacks novelty, as it merely confirms responses previously observed in many other EMF species. The findings of this manuscript are simple, insufficient and does not bring valuable information.

1.Title of the MS is quite inappropriate. Suggest change the title to “The physiological Response of Ectomycorrhizal fungus Lepista sordida to Cd and Cu stress”.

2.Line 117, It is not clear how to measure the dry weight of mycelia by SEM? please add the related details.

3.It is so easy to measure the levels of Cu and Cd in. Why not to determine metal concentration of mycelia.

4.For different analyses, authors used different time (10 d, 25 d). What was the reason for choosing two harvesting periods?

5.The caption of Figures 3, 4, 6 should be modified as the image, and more concise.

6.The legends of Fig. 8, 9 should be modified as the image

7.Check reference:I noticed few unnecessary abbreviation of author names there (Lines 60-63, 80,82)

8.This article doesn't concern the Proline in the plant tissues, please correct the sentence of Line 157.

9.In discussion section, the authors repeated some results again, but without any further discussion.

10.“when the concentration of Cu increased from 0.1 to 0.5 mmol L-1, the decrease in the colony diameter was not significant (Fig.4)”. However the mycelial dry weight under Cd stress was significantly different from that in the control group in Fig.7B, how to explain this difference?

Annotated reviews are not available for download in order to protect the identity of reviewers who chose to remain anonymous.

---

## Round 0.3 · accepted · Accept

This version is accepted for publication.